



# Induced telluric currents play a major role in the interpretation of geomagnetic variations

Liisa Juusola[1], Heikki Vanhamäki[2], Ari Viljanen[1], and Maxim Smirnov[3]

[1]Finnish Meteorological Institute, Helsinki, Finland
[2]University of Oulu, Oulu, Finland
[3]Luleå University of Technology, Sweden

**Correspondence:** L. Juusola (liisa.juusola@fmi.fi)

**Abstract.** Geomagnetically induced currents (GIC) are directly described by ground electric fields, but estimating them is time-consuming and requires knowledge of the ionospheric currents as well as the three-dimensional distribution of the electrical conductivity of the Earth. The time derivative of the horizontal component of the ground magnetic field ($dH/dt$) is closely related to the electric field via Faraday's law, and provides a convenient proxy for the GIC risk. However, forecasting $dH/dt$
still remains a challenge. We use 25 years of 10 s data from the North European International Monitor for Auroral Geomagnetic Effects (IMAGE) magnetometer network to show that part of this problem stems from the fact that instead of the primary ionospheric currents, the measured $dH/dt$ is dominated by the signature from the secondary induced telluric currents nearly at all IMAGE stations. The largest effects due to telluric currents occur at coastal sites close to highly-conducting ocean water and close to near-surface conductivity anomalies. The secondary magnetic field contribution to the total field is a few tens of
percent, in accordance with earlier studies. Our results have been derived using IMAGE data and are thus only valid for the involved stations. However, it is likely that the main principle also applies to other areas. Consequently, it is recommended that the field separation into internal (telluric) and external (ionospheric and magnetospheric) parts is performed whenever feasible, i.e., a dense observation network is available.

# 1 Introduction

Fast geomagnetic variations at periods from seconds to hours and days are primarily produced by currents in the ionosphere and magnetosphere. There is always an associated secondary (internal, telluric) current system induced in the conducting ground and contributing to the total variation field measured by ground magnetometers. Mathematically, it is possible to fully explain the variation field by two equivalent current systems, one at the ionospheric altitude and another just below the Earth's
surface. In practice, this separation is feasible using dense magnetometer networks (Pulkkinen et al., 2003b; Stening et al., 2008; Juusola et al., 2016). A common way in space physics has been to implicitly neglect the internal part and interpret the ground field only in terms of ionospheric (and magnetospheric) equivalent currents. As known from previous studies (Viljanen



et al., 1995; Pulkkinen and Engels, 2005; Pulkkinen et al., 2006), this is often a reasonable assumption, since a typical internal contribution is about 10–30%.

Geomagnetically induced currents (GIC, Boteler et al., 1998) in long technological conductor systems, such as power grids, are a significant space weather concern. They are directly described by ground electric fields, which are associated with the time derivative of the magnetic field via Faraday's law. The time derivative of the horizontal ground magnetic field ("$dH/dt$") can be used as a proxy for the GIC risk level (Viljanen et al., 2001). Auroral substorms are one of the major causes of large $dH/dt$ values (Viljanen et al., 2006). During substorm onsets, the internal contribution to the ground magnetic field can be up to

40% (Tanskanen et al., 2001). However, there seems to be very little previous information on how much telluric currents affect $dH/dt$. Understanding of the effects of telluric currents on $dH/dt$ is also relevant when models' ability to forecast ground magnetic perturbations is validated by comparing them with measurements (Pulkkinen et al., 2013; Welling et al., 2018).

Geomagnetic induction is a complicated phenomenon with intricate dependencies between the scale sizes of the ground conductivity structures and the spatiotemporal composition of the ionospheric primary fields. A widely-used simplification in

the frequency domain is to consider the effects of a primary plane wave field on a one-dimensional (1-D, i.e., variation as a function of depth only) electrical conductivity distribution of the Earth. In such a case, the contribution of the secondary field is 50% (Tanskanen et al., 2001, Eq. 4) for both $H$ and $dH/dt$. In reality, the conductivity distribution is three-dimensional (3-D) and the primary field is not a plane wave.

A well-known example of the strong influence of the 3-D conductivity distribution is the so-called "coast effect" (Parkinson,

1959; Rikitake and Honkura, 1985). It is caused by the conductivity contrast between the well-conducting sea water and adjacent land area. The coast effect is a two-fold phenomenon. First, the effect is observed as a large amplitude of the ratio of the vertical magnetic field component to the horizontal component (tipper vector, induction arrow) at a particular frequency. This is caused by the concentration of the induced current density in the well-conducting sea, which produces a vertical magnetic field at its edge (sea-land interface), resulting in the steepening of the observed fields (Transverse Electric or TE-

mode). Second, because the induced currents normal to the sea-land interface are continuous, electric fields are discontinuous, and strongly amplified on the land side (Transverse Magnetic or TM-mode). It should be noted that this "coast effect" can be observed at any large conductivity contrast, such that electric fields and vertical magnetic fields are amplified above the less-conducting region. Depending on the geometry of the power grid, the enhanced electric field can increase GIC near coasts or large conductivity anomalies. Parkinson and Jones (1979) noticed that in addition to induction in the sea, similar effects may

be caused by conductivity contrasts in the deeper (mantle) structure between continent and ocean.

The non-plane wave primary field together with the effects of the 3-D conductivity distribution typically reduce the secondary contribution to $H$ compared to a plane-wave field and 1-D conductivity. It is not self-evident that the effect of realistic induction on $dH/dt$ would be similar to that on $H$, because the frequencies ($\omega$) of the time-varying field dominating the observed $H$ and $dH/dt$ signatures are not expected to be the same: if the Fourier transform of $H(t)$ is $h(\omega)$, then the Fourier transform

of $dH/dt$ is $\omega h(\omega)$, indicating that higher frequencies are more pronounced in $dH/dt$ than in $H$ (i.e., the time derivative acts as a high-pass filter). Because the measured $H$ is a sum of the primary and secondary $H$, and the secondary $H$ is driven by



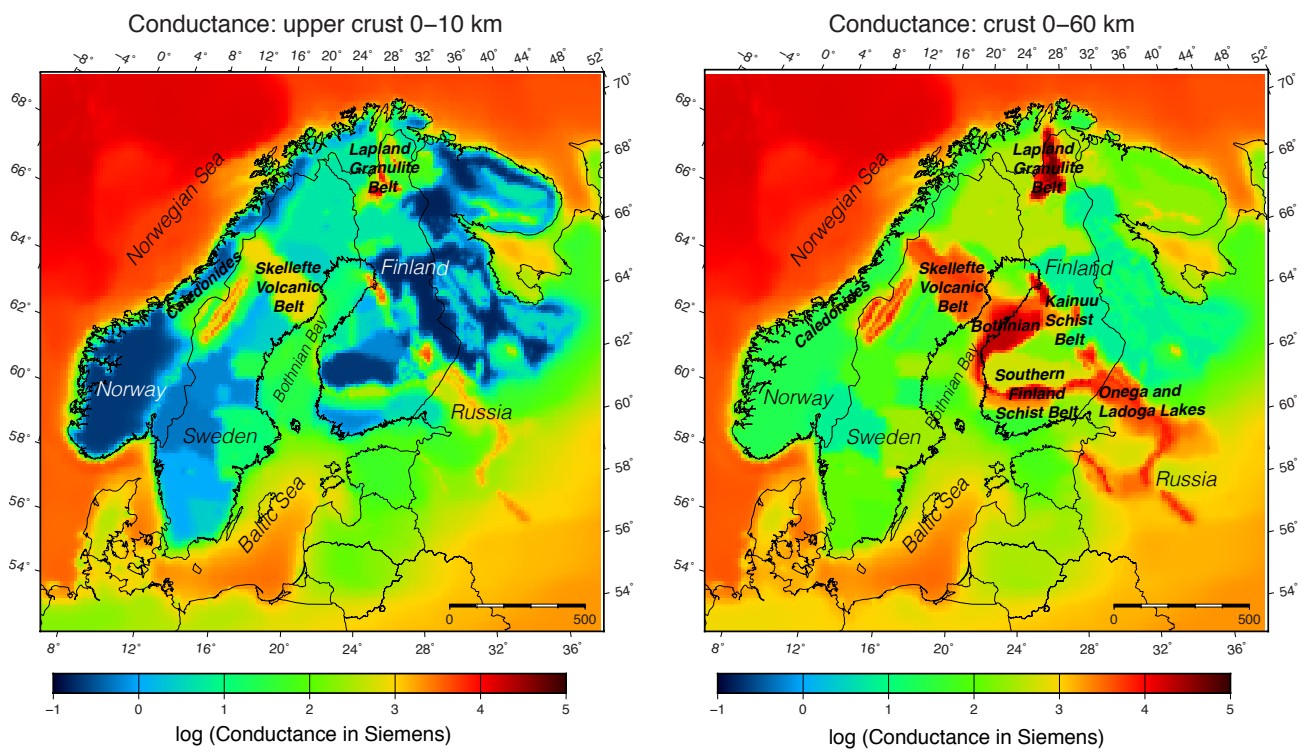

**Figure 1.** Conductance of the upper crust (0–10 km) and crust (0–60 km) based on SMAP data (Korja et al., 2002).

the primary $dH/dt$, induction amplifies higher frequencies present in the primary $H$ (and $dH/dt$) more strongly than lower frequencies.

There are two key factors that determine the distribution of the telluric current density and, thus, the secondary induced

magnetic field. One is the time varying external magnetic field that drives the induction. The main origin of this primary field is the ionospheric current density, with some contribution from the more distant magnetospheric currents. The other factor is the Earth's conductivity distribution. A conductance map of the Fennoscandian Shield and its surrounding oceans, sea basins, and continental areas (SMAP) has been presented by Korja et al. (2002), based on information from deep electromagnetic geophysics (magnetotellurics) and geology. We have used SMAP data to illustrate the conductances at 0–10 km depth and

0–60 km depth. These are presented in Fig. 1.

Key features of the conductivity model relevant for telluric currents are the well-conducting seawater and sea sediments surrounding the Fennoscandian Shield, which consists of a highly resistive crust with imbedded well-conducting belts. Engels





et al. (2002) used SMAP together with a primary plane wave magnetic field to model the telluric currents in the frequency domain (period of 2048 s ≈34 min). According to their results, the majority of the induced current was concentrated in the seawater and conductivity anomalies. There were prominent effects due to strong electrical conductivity contrasts around coast lines and conductivity anomalies.

The International Monitor for Auroral Geomagnetic Effects (IMAGE, https://space.fmi.fi/image/www/) magnetometer network covers the same area as the map of Korja et al. (2002). The detailed information on the crustal conductivity combined with the long time series of magnetic field observations provide an excellent opportunity to study the effects of telluric currents on the ground magnetic field and its time derivative in this area. We use 10 s magnetic field data measured by IMAGE in 1994–2018 and separate the data into internal (induced telluric) and external (driving ionospheric-magnetospheric) parts using the Spherical Elementary Current System (SECS, Vanhamäki and Juusola, 2020) method. Each time step is processed independently of the others and no assumptions about the ground or ionospheric conductivity are made, except that there can be induced currents at any depth below the Earth's surface and that there are no electric currents between the ground and 90 km altitude. This data set is used to carry out to our knowledge the first extensive statistical analysis on the effects of 3-D induction on the ground magnetic field and, especially, its time derivative. The results are interpreted in the light of our knowledge of the underlying ground conductivity (Korja et al., 2002; Engels et al., 2002).

The Earth's conductivity distribution is occasionally considered to consist of two components: a normal 1-D component and an anomalous 3-D component. Similarly, the induced field is considered to consist of a normal part and an anomalous or scattered part. We have not made this separation but consider the normal and anomalous parts together. Unless otherwise mentioned, all analysis in this study is carried out in the time domain, i.e., by considering time series.

## 2 Data and method

### 2.1 Data

We use 10 s ground magnetic field measurements from the International Monitor for Auroral Geomagnetic Effects (IMAGE, https://space.fmi.fi/image/www/) magnetometers in 1994–2018. Currently, IMAGE consists of 41 stations that cover magnetic latitudes from the subauroral 47° N to the polar 75° N in an approximately two hour magnetic local time (MLT) sector.

### 2.2 Method

Because most IMAGE stations are variometers, we cannot use a model to subtract the baseline from the data. Instead, we have used the method by van de Kamp (2013) to remove the long-term baseline (including instrument drifts, etc.), any jumps in the data, and the diurnal variation. The diurnal quiet-time magnetic field variation in the IMAGE region is at most a few tens of nT (Sillanpää et al., 2004). We concentrate on studying large time derivatives of the horizontal magnetic field for which this effect is insignificant.





After the baseline subtraction, we have applied the two-dimensional (2-D) Spherical Elementary Current System (SECS) method (Amm, 1997; Amm and Viljanen, 1999; Pulkkinen et al., 2003a, b; Juusola et al., 2016; Vanhamäki and Juusola, 2020)
to calculate the ionospheric and telluric current densities for each time step and to separate the magnetic field measured at each station into internal and external parts. To make sure that all currents in space flow beyond the ionospheric equivalent current sheet and all telluric currents below the telluric equivalent current sheet, we place these sheets at 90 km altitude and 1 m depth, respectively. The actual depth distribution of the currents cannot therefore readily be concluded from this analysis. Pulkkinen et al. (2003b) set the internal layer at the depth of 30 km, but such a choice omits induced currents close to the Earth's surface.

A change in the station configuration can, under certain conditions, result in an artificial time derivative peak in the separated magnetic field at the nearby stations. Because of this, we have discarded any station with data gaps during a day. The time derivative has been calculated so that values during succesive days are not compared. This is a fairly strict approach and wastes some usable data, but ensures that there will not be any artificial time derivative peaks due to changes in station configuration.

IMAGE data are provided in geographic coordinates and we carry out the analysis using the same coordinate system. We
use the notations $B_x$, $B_y$, and $B_z$ for the north, east and down components of the ground magnetic field. The horizontal magnetic field vector is denoted by $\boldsymbol{H} = B_x \hat{\boldsymbol{e}}_x + B_y \hat{\boldsymbol{e}}_y$ and its amplitude by $H = \sqrt{B_x^2 + B_y^2}$. Similarly, the time derivative vector and its amplitude are $d\boldsymbol{H}/dt = dB_x/dt\,\hat{\boldsymbol{e}}_x + dB_y/dt\,\hat{\boldsymbol{e}}_y$ and $dH/dt = \sqrt{(dB_x/dt)^2 + (dB_y/dt)^2}$, respectively. The measured magnetic field is a sum of the telluric and ionospheric contributions, e.g., $B_x = B_{x,telluric} + B_{x,ionospheric}$. Although geographic coordinates are used to present the data, we have occasionally marked the magnetic coordinates in the plots. We
have used the quasi-dipole (QD) coordinates (Richmond, 1995; Emmert et al., 2010) as given by the software available at https://apexpy.readthedocs.io/en/latest/. The code uses the 12th generation International Geomagnetic Reference Field (IGRF-12, Thébault et al., 2015).

## 3 Results

### 3.1 Example event

Figure 2 shows an example of the ionospheric (2a) and telluric (2b) equivalent current densities and their time derivatives (2c–d) on 18 March 2018 at 21:22:30 UT. The arrows illustrate the vector quantity and the color shows the corresponding horizontal component of the ground magnetic field. Magnetic latitude and magnetic local time are indicated by the blue grid. The locations of the IMAGE stations used to construct the maps are shown with black squares, and station SOD is highlighted with a thicker marker line. The black, vertical line passing through SOD indicates the meridian along which the horizontal
ground magnetic field has been extracted in order to construct Figure 3.

The telluric current density and its time derivative are mainly directed opposite to the driving ionospheric current density and its time derivative, as expected. However, whereas the ionospheric currents are clearly oblivious to the conductivity structure of the Earth, the telluric currents are strongly affected by it. The peak of the telluric current density does not coincide with the peak of the westward electrojet but is displaced northward, favoring the highly conducting sea area over the more resistive land
area. The difference in the driving and induced patterns illustrates clearly the coast effect where the current flowing in the sea



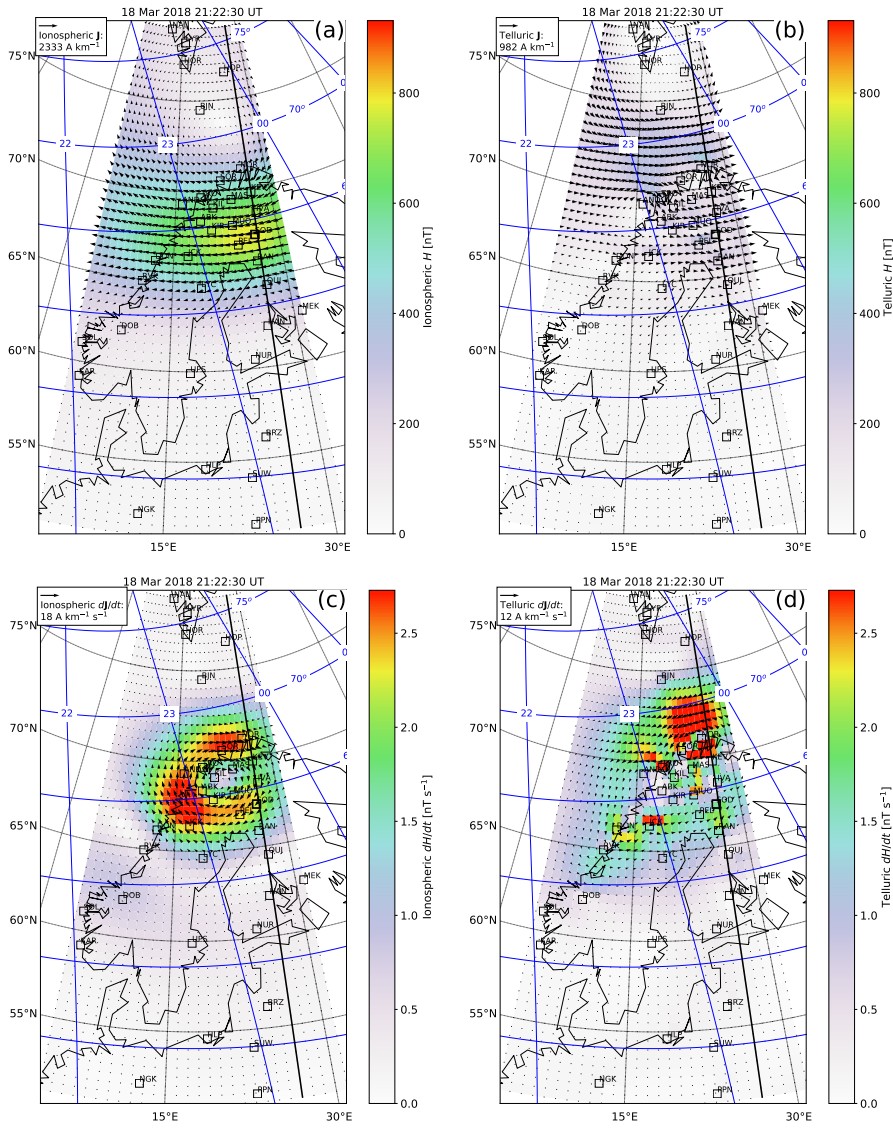

**Figure 2.** Ionospheric equivalent current density (arrows) on 18 March 2018 at 21:22:30 UT (a), derived from IMAGE magnetic field measurement. The color shows the corresponding horizontal component of the ground magnetic field. Magnetic latitude and magnetic local time are indicated by the blue grid. Locations of the IMAGE stations are shown with black squares, and station SOD is highlighted by a thicker marker line. The black, vertical line passing through SOD indicates the meridian along which the horizontal ground magnetic field has been extracted in order to construct Figure 3. (b): Telluric equivalent current density and corresponding ground magnetic field. (c): Time derivative of the ionospheric equivalent current density and corresponding time derivative of the horizontal ground magnetic field. (d): Time derivative of the telluric equivalent current density and corresponding time derivative of the horizontal ground magnetic field.





area encounters the highly resistive crust of the land area. The presence of highly conducting elongated structures within the land area (Korja et al., 2002) is also evident in the induced currents. This behavior is in agreement with the modeling results by Engels et al. (2002), performed in the frequency domain with the plane wave assumption and 3-D conductivity distribution. The amplitude of the horizontal ground magnetic field due to telluric currents (Fig. 2b) is clearly weaker than that due to the
ionospheric currents (Fig. 2a). However, the telluric and ionospheric contribution to the time derivative of the magnetic field are of comparable strength.

The time development of the event surrounding the above example is illustrated in Fig. 3 and in the animation provided as supplementary material. Fig. 3a shows the local IMAGE equivalents of the auroral electrojet indices (Davis and Sugiura, 1966; Kauristie et al., 1996), called IL and IU, as thick and thin black curves, respectively. The corresponding values derived from
the ionospheric and telluric parts of the separated magnetic field are plotted in blue and red. The rest of the panels show time series of latitude profiles of the ionospheric (3b) and telluric (3c) contributions to the horizontal ground magnetic field and their time derivatives (3d–e) along the longitude of SOD (black, vertical lines in Fig. 2a–d). The time interval shown in Fig. 3 is 21:00:00–22:00:00 UT and the time of the example in Fig. 2 is marked with the black, vertical line. The animation consists of a time series of frames showing plots similar to Fig. 3 and Fig. 2 from 21:00:00–22:00:00 UT with a 10 s time step.

The event consists of an intensification and subsequent decay of a westward electrojet (Fig. 3a) around the magnetic mid-night. The example in Fig. 2 took place during the intensification, when the largest time derivatives (Fig. 3d–e) were observed at SOD (MLT ≈ UT+2.5 h). While the equivalent currents and ground magnetic fields change quite slowly in time and space, their time derivatives are highly dynamic. Although the ionospheric time derivative structures only live some tens of seconds, in agreement with Pulkkinen et al. (2006), they still display fairly smooth structure and time development. The telluric
time derivative structures in the land area, on the other hand, are spatially much more variable because of the complex 3-D conductivity distribution.

Figure 4a–c shows the measured magnetic field components (black) as well as their ionospheric (blue) and telluric contributions (red) at SOD. As expected, the telluric currents strengthen the ionospheric $B_x$ by a few tens of percent (Viljanen et al., 1995), while the ionospheric and telluric $B_z$ are oppositely directed. For this event, $B_y$ is relatively weak, as expected for a
westward electrojet. Fig. 4d–f shows the time derivative of the magnetic field. Unlike the horizontal magnetic field components, the time derivatives of $B_x$ and $B_y$ are mostly dominated by the telluric component.

In order to examine what are the relevant periods for the ionospheric and telluric magnetic fields and their time derivatives, we perform wavelet transforms (e.g., Torrence and Compo, 1998; Fligge et al., 1999) on the measured, ionospheric, and telluric $B_x$ and $dB_x/dt$. We use continuous wavelet transform with Morlet wavelets as given by the software available at
https://pywavelets.readthedocs.io/en/latest/ (Gregory et al., 2019). The results are shown in Fig. 5. Note that the periodic structures visible in the plots are artificial. They are caused by the definition of the wavelets used and would be different for other wavelets. In addition to the one hour interval shown in Fig. 4a and Fig. 4d, we have included one hour of data before and after the interval of interest, i.e., analyzed a three hour interval, but limited the periods shown in Fig. 5 to 1 h. The black, vertical line in Fig. 5 indicates the time shown in Fig. 2. The period ranges of the ultra-low frequency (ULF) pulsation classes
Pc4 (45–150 s) and Pc5 (150–600 s) (Jacobs et al., 1964) are shown with the white, horizontal, dashed lines.



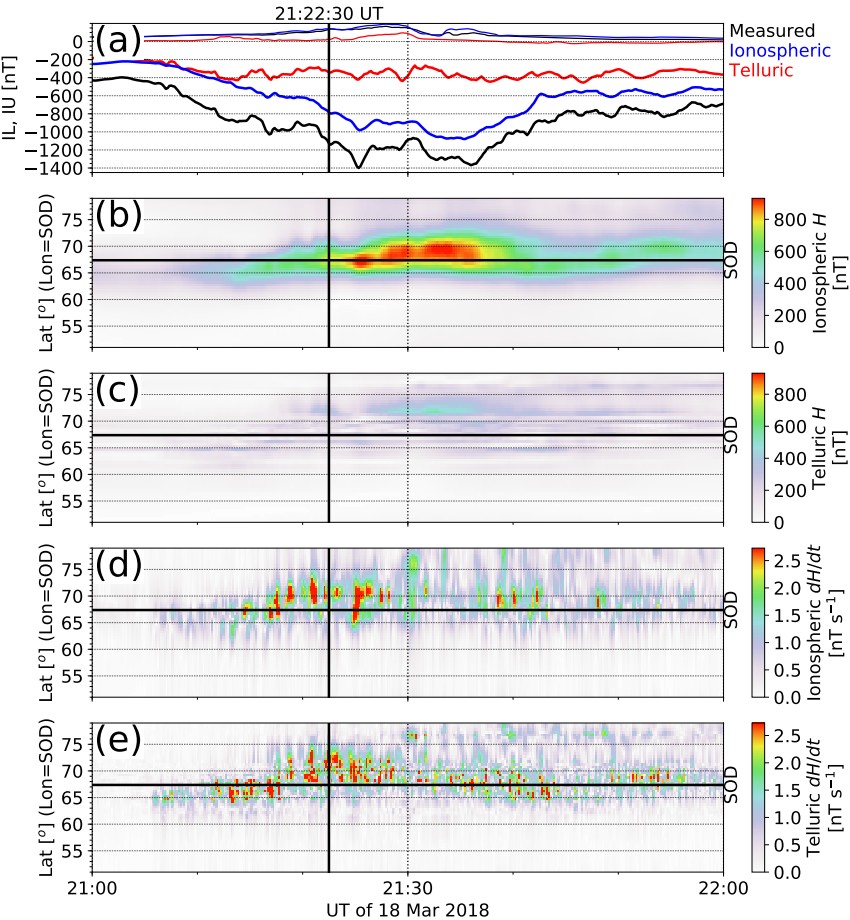

**Figure 3.** Upper (IU, thin black curve) and lower (IL, thick black curve) envelope curves of the magnetic field $x$ component measured by IMAGE as a function of UT on 18 March 2018 at 21:00:00–22:00:00 UT. IL and IU derived from the separated ionospheric and telluric parts of the magnetic field are shown in blue and red, respectively (a). Latitude profiles of ionospheric contribution to horizontal ground magnetic field (b), telluric contribution to horizontal ground magnetic field (c), ionospheric contribution to the time derivative of the horizontal ground magnetic field (d), and telluric contribution to the time derivative of the horizontal ground magnetic field (e) along the longitude of SOD as a function of UT. The black, horizontal line indicates the latitude of SOD, and the black, vertical line indicates the time shown in Fig. 2.

While most of the measured (Fig. 5a) and ionospheric $B_x$ (Fig. 5b) signals consist of longer periods above the Pc5 threshold of 600 s, the shorter periods in the Pc5 range are somewhat more relevant for the telluric $B_x$ (Fig. 5c), and clearly more relevant for the measured (Fig. 5d) and ionospheric $dB_x/dt$ (Fig. 5e). For the telluric $dB_x/dt$ (Fig. 5f) signal, on the other hand, periods in the Pc5 and even Pc4 range are very significant, with only some contribution from the longer periods. This behavior is in agreement with our discussion on the relevant frequencies in the Introduction. It can be seen as well that changes in the ionospheric $B_x$ power (Fig. 5b) at a certain frequency are associated with intensifications in the ionospheric $dB_x/dt$ power (Fig. 5e), as expected. However, comparing the power of ionospheric (Fig. 5e) and telluric $dB_x/dt$ (Fig. 5f) at the Pc5





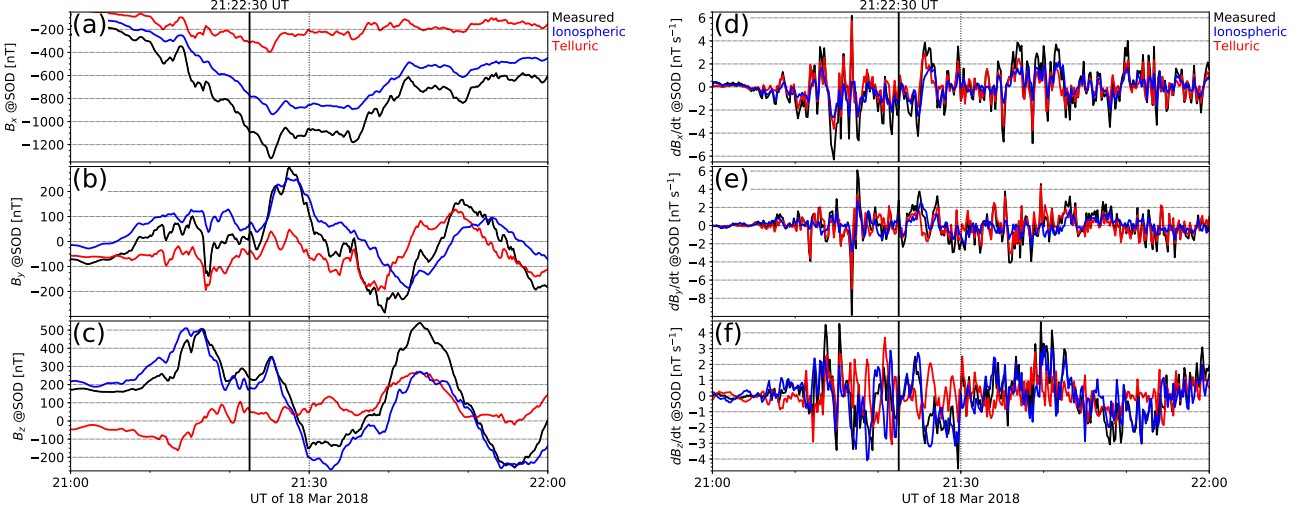

**Figure 4.** Magnetic field north component $B_x$ (a) and its time derivative $dB_x/dt$ (d), east component $B_y$ and its time derivative $dB_y/dt$ (b,e), and down component $B_z$ and its time derivative (c,f) at SOD as a function of UT for the event in Fig. 2. The measured value is plotted in black, the ionospheric contribution in blue, and the telluric contribution in red. The black, vertical line indicates the time shown in Fig. 2.

band around 21:15 UT and 21:25 UT, it can be seen that the ratio of the ionospheric and telluric contributions is not constant. Rather, it must depend on the spatiotemporal structure of the ionospheric current system. The telluric $B_x$ power tends to more

or less follow the behavior of the ionospheric $dB_x/dt$ with a small delay. This delay is a consequence of induction in a realistic earth with a finite conductivity, and will be discussed further in Sect. 4.

## 3.2 Telluric contribution to $H$ and $dH/dt$ at SOD

In order to further examine the relative contributions of ionospheric and telluric currents to the horizontal components of the ground magnetic field and their time derivatives, Fig. 6 shows the telluric contribution to $B_x$, $B_y$, and their time derivatives

as a function of the measured value at SOD in 1996–2018. Only values with large time derivatives of the horizontal magnetic field ($dH/dt = \sqrt{(dB_x/dt)^2 + (dB_y/dt)^2} > 1\,\mathrm{nTs}^{-1}$) (Viljanen et al., 2001) are included to concentrate on time steps when large GIC are most likely to occur. This is roughly 1% of the total number of data points. The black line in Fig. 6 is the line of unity and the red line is a least squares fit to the data points. The slope of this line is given in the top right corner of the panel, indicating a typical telluric contribution of 29% to $B_x$, 46% to $B_y$, 54% to $dB_x/dt$, and 65% to $dB_y/dt$. While the

telluric contribution to $B_x$ is fairly modest and in agreement with earlier results (Viljanen et al., 1995; Tanskanen et al., 2001; Pulkkinen and Engels, 2005; Pulkkinen et al., 2006), the other contributions, especially those to the time derivatives, are quite high.



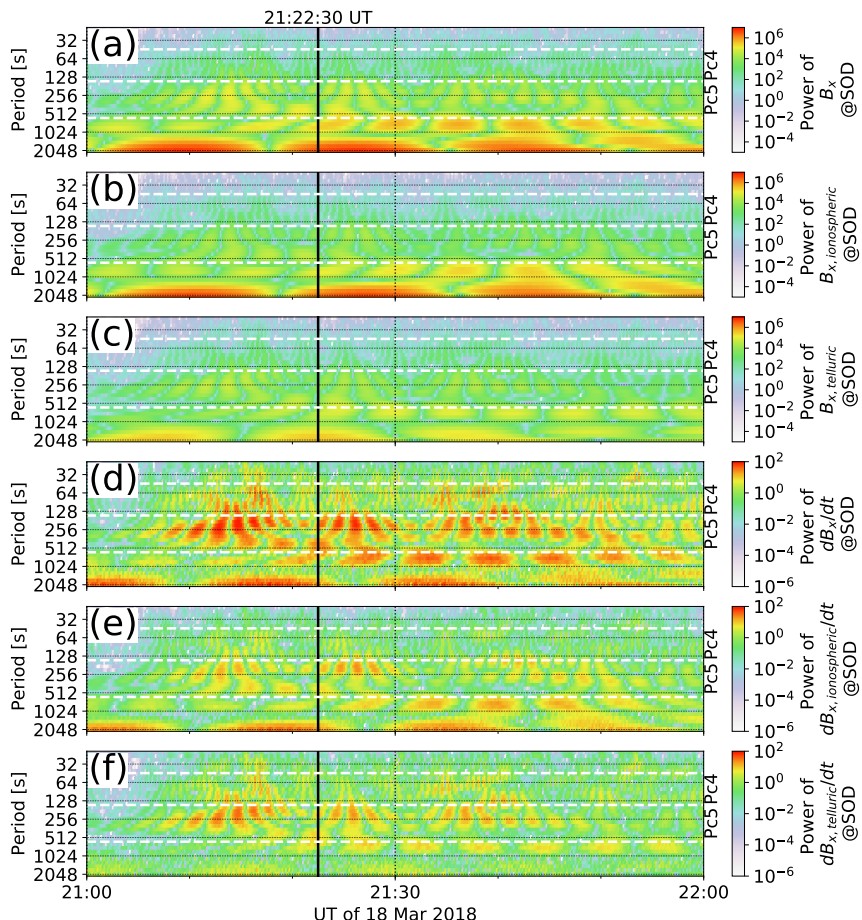

**Figure 5.** Wavelet transform of the magnetic field north component $B_x$ at SOD as a function of UT for the event in Fig. 2 (a). The same for the primary ionospheric (b) and secondary telluric $B_x$ (c), and their time derivatives (d–e). The black, vertical line indicates the time shown in Fig. 2. The period ranges of the ultra-low frequency (ULF) pulsation classes Pc4 (45–150 s) and Pc5 (150–600 s) (Jacobs et al., 1964) are shown with the white, horizontal, dashed lines.

### 3.3 Telluric contribution to $H$ and $dH/dt$ at IMAGE stations

So far we have concentrated on one IMAGE station only. We will now extend the analysis to the rest of the stations available in 1994–2018. The station LOZ has been omitted from the analysis because the data showed some unphysical behavior, and the newest IMAGE stations RST, HAR, BRZ, HLP, SUW, WNG, NGK, and PPN, because there was not enough data available from them to produce reliable statistics. In this section we have again only considered measurements that have large horizontal time derivatives, i.e., $dH/dt > 1\,\mathrm{nTs}^{-1}$. The number of such data points for each station is listed in Table 1.





**Figure 6.** Telluric contribution to $B_x$ as a function of measured $B_x$ at SOD in 1996–2018 (a). Only values with large time derivatives of the horizontal magnetic field ($dH/dt > 1$ nTs$^{-1}$) are included. The black line is the line of unity and the red line is a least squares fit to the data points. The slope of this line is indicated in the top right corner of the panel. The same for $B_y$ (b), $dB_x/dt$ (c), and $dB_y/dt$ (d).

Fig. 7a shows the slope $k$ of the fitted line $B_{x,telluric} = k \cdot B_x + const.$ for each IMAGE station. The same for $B_y$, $dB_x/dt$,
and $dB_y/dt$ are shown in Fig. 7b–d. Magnetic coordinates are indicated by the blue grid with the separation of the constant latitude lines corresponding to one hour in MLT. Numerical values of $k$ are listed in Table 1.

The smallest induced contribution can be observed at stations KIL, ABK, MUO, and KIR. These stations are 1) typically located below the driving ionospheric currents. The internal contribution tends to increase away from the main ionospheric current system (Pulkkinen and Engels, 2005), which is also visible in the simplified model applied by Boteler et al. (1998).
This effect is probably at least partly responsible for the larger telluric contribution at the more southern IMAGE stations. For a 1-D Earth and a plane wave primary field, the secondary contribution would be 50%. 2) Located away from the coastline.



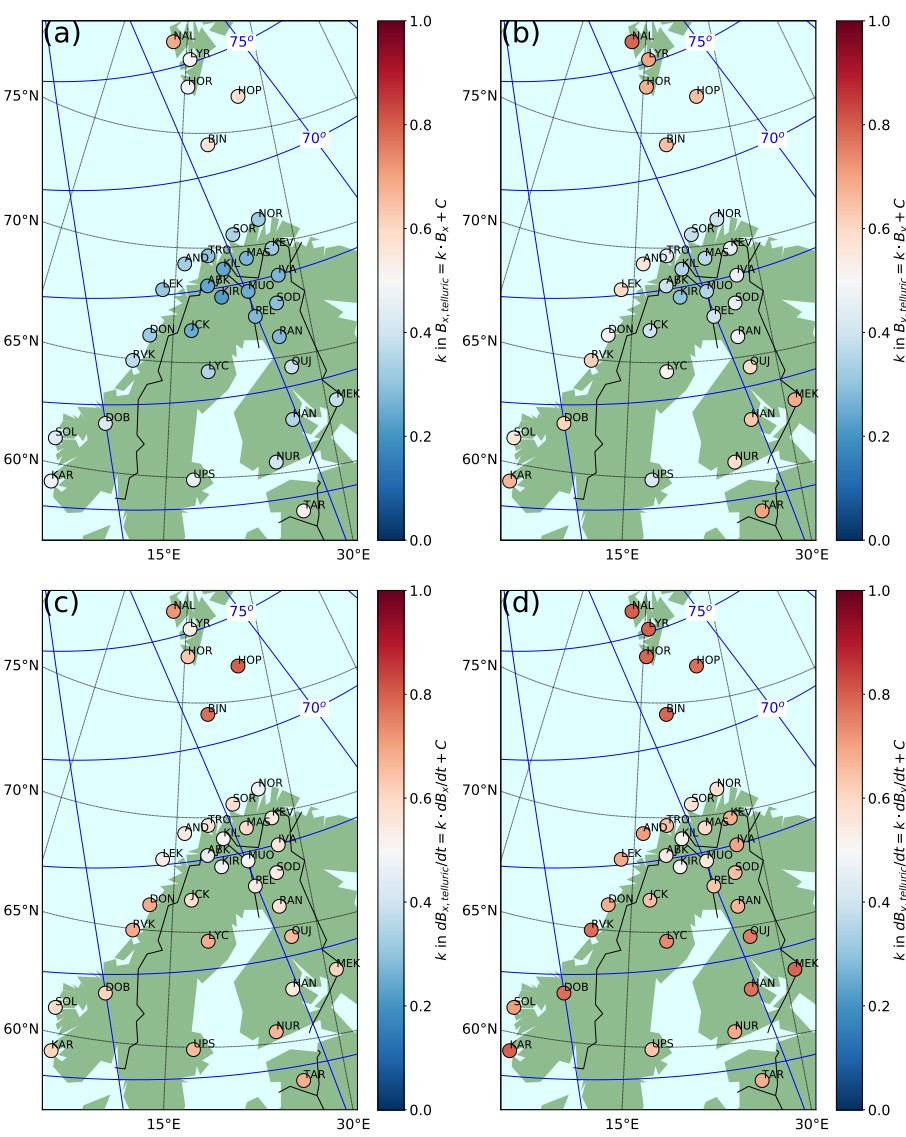

**Figure 7.** Slope $k$ of the fitted line $B_{x,telluric} = k \cdot B_x + const.$ with $dH/dt > 1$ nTs$^{-1}$ for IMAGE stations with sufficient amounts of good data available in 1994–2018 (a). The same for $B_y$ (b), $dB_x/dt$ (c), and $dB_y/dt$ (d). Magnetic coordinates are indicated by the blue grid. The separation of the arbitrarily placed constant longitude lines corresponds to one hour in MLT.

There is a clear increase in the internal contribution to $B_y$ and $B_x$ at the Norwegian coastal stations, due to the typical primary ionospheric currents flowing in the east-west direction and the secondary induced currents turning to follow the coastline. 3) Located away from the conductivity anomalies on land. There are two prominent conductivity anomalies (see Fig. 1): one related to the Archean-Proterozoic boundary (Hjelt et al., 2006) and directed approximately from northwest to southeast, affecting $B_x$ and $B_y$ at least at RVK, DON, LYC, OUJ, and MEK. The other conductivity structure is directed from north to






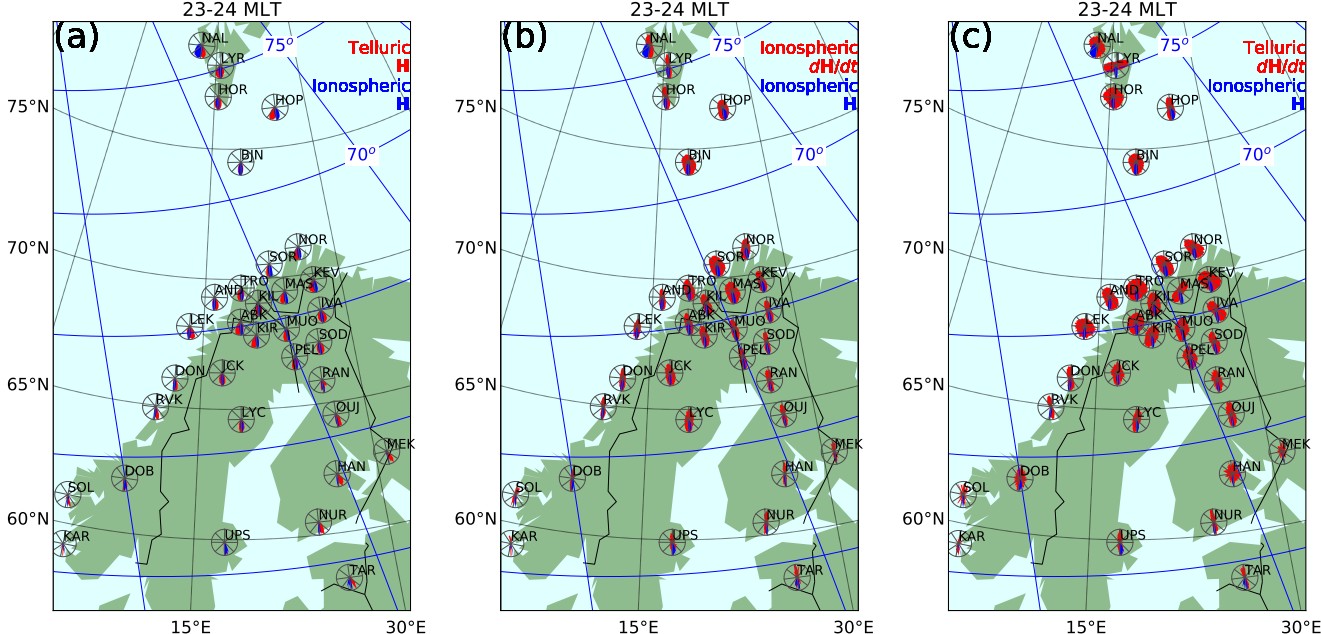

**Figure 8.** Histograms of the direction of the ionospheric (blue) and telluric (red) horizontal ground magnetic field when $dH/dt > 1\,\mathrm{nTs}^{-1}$ and $23\,\mathrm{h} \leq \mathrm{MLT} < 24\,\mathrm{h}$ for IMAGE stations with sufficient amounts of good data available in 1994–2018 (a). In (b) the telluric part of the horizontal magnetic field has been replaced by the time derivative of the ionospheric part of the ground magnetic field and in (c) with the time derivative of the telluric part of the ground magnetic field.

south and affects $B_y$ at least at KEV, IVA, and SOD. It should be noted that recent studies (Cherevatova et al., 2015) indicate much more complex structure of the above mentioned conductivity anomalies.

Finally, we will examine the effect of the field separation on the direction of the horizontal ground magnetic field vectors and their time derivatives at the IMAGE stations. Because the typical direction of the field is strongly dependent on MLT, we have divided the data into one hour MLT bins. Figure 8 shows the results for the 23–24 h MLT bin. Plots for the other MLTs are provided as supplementary material as well as a Table listing the number of data points in each bin. Fig. 8a shows histograms of the direction of the telluric (red) and ionospheric (blue) contribution to $\boldsymbol{H}$. The blue histograms in Fig. 8b–c are the same as in 8a, but the red histograms illustrate the direction of the ionospheric (8b) and telluric (8c) contribution to the time derivative

vector $d\boldsymbol{H}/dt$.

The telluric $\boldsymbol{H}$ is typically more or less in the same direction as the ionospheric $\boldsymbol{H}$ and none of the stations stands out by behaving radically different from other nearby stations. The number of data points decreases southward and consequently the histograms of the southern IMAGE stations are clearly more noisy than those of the northern stations. Because large time derivatives tend to occur around midnight and morning hours (Viljanen et al., 2001), the histograms for all stations tend to be

relatively noisy at other times.





The ionospheric $d\boldsymbol{H}/dt$ also tends to be more or less aligned with the ionospheric $\boldsymbol{H}$, except at auroral latitudes during morning hours when the ionospheric $d\boldsymbol{H}/dt$ tends to be more strongly east-west directed than the ionospheric $\boldsymbol{H}$. This behavior is in agreement with Viljanen et al. (2001).

The telluric $d\boldsymbol{H}/dt$ histograms tend to be wider than the ionospheric ones. They also reveal some clear anomalies. The most pronounced ones are at MAS and LYR, where the telluric $d\boldsymbol{H}/dt$ has a preferred direction that at many MLTs differs markedly from those of the ionospheric $\boldsymbol{H}$ and $d\boldsymbol{H}/dt$. At the coastal stations RVK, DON, AND, LEK, TRO, SOR, NOR, the telluric $d\boldsymbol{H}/dt$ shows a preference to a direction perpendicular to the local coast line, most likely because of strong induced currents flowing along the coast. At IVA, KEV, and SOD, the telluric $d\boldsymbol{H}/dt$ tends to prefer a more east-west aligned direction than the driving ionospheric field. This is most likely due to the local north-south aligned conducting belt. Viljanen et al. (2001) list AND, LYC, MAS, and TRO as stations where the directional distribution of the measured $d\boldsymbol{H}/dt$ is strongly rotated or scattered by telluric currents. Examination of the MLT dependency of the telluric $d\boldsymbol{H}/dt$ at LYC shows that the presence of the nearby northwest-southeast aligned conducting belt tends to rotate the vectors accordingly.

## 4 Discussion

We have used 10 s magnetic field measurements from the IMAGE network in 1994–2018 to demonstrate that although the telluric contribution to the measured magnetic field is modest, as expected based on earlier studies, the contribution to the time derivative is significant. The separation of the measured magnetic field into internal and external parts was carried out using the 2-D SECS method. Each time step was processed independently of the others and no assumptions about the ground or ionospheric conductivity structure were made, except that there can be induced currents at any depth below the Earth's surface and that there are no electric currents between the ground and 90 km altitude. The relations between the internal and external field components can be well explained by the known major conductivity structures (Korja et al., 2002).

### 4.1 Suggested explanation

Although the significance of the telluric currents to the time derivative has according to our knowledge not been considered until now, the qualitative explanation is quite straightforward. It is well known that the electromagnetic field penetrates into the Earth in a diffusive manner. The penetration depth depends on the subsurface conductivity ($\sigma$) and period ($T$) of the electromagnetic field, as described by the skin depth $s = \sqrt{\sigma^{-1}T}$. Thus, faster variations have a shallower penetration depth.

Penetration depth does not directly describe the depth of the induced current, which creates the telluric part of the magnetic field, but the depth by which the inducing field has lost most of its energy. Thus, the majority of the induced current should flow above the penetration depth. Significant induced current density can be produced if there is a sufficiently sized structure of sufficiently good conductivity at a suitable depth considering the period of the inducing field and the conductance structure through which it needs to diffuse to reach that structure.

Generally, conductivity is very low at the Earth's surface and increases with depth. Hence, the slower variations that dominate the ionospheric part of $H$ would be expected to induce currents that are stronger (relative to the primary wave energy) and





located deeper than those induced by the faster variations that dominate the ionospheric part of $dH/dt$. However, the highly
conducting sea and near-surface conductivity anomalies change the picture dramatically. The conductivity anomalies are typi-
cally not large enough to catch the slower variations, and the sea, although it can cover large areas, is most likely too shallow
to catch a very large portion of the wave energy. For the faster variation, on the other hand, the sea and the anomalies are very
good conductors at an optimal depth, catching the majority of the wave energy. Thus, in a realistic 3-D earth, faster magnetic
field variations would be expected to induce stronger (relative to the primary wave energy) currents closer to the surface than
slower variations. Thus, the earth would be expected to amplify ground $dH/dt$ more strongly than $H$.

### 4.2 Simple models vs. reality

The simplest model to explain the effect of the telluric currents is to assume a perfect conductor at some depth in the earth
(e.g., Pulkkinen et al., 2003b; Kuvshinov, 2008), or in a special case, a 2-D structure is also possible (Janhunen and Viljanen,
1991). Such models give qualitative understanding of the internal contribution to the magnetic field, but they can be very
misleading when applied to the time derivative. For simplicity, consider planar geometry with a perfect conductor. Then the
induced currents could be replaced by mirror images of the external currents. The induced fields follow strictly the temporal
behavior of the external currents. Then the relative internal contribution at a given location is the same for both the magnetic
field and its time derivative.

Contrary to this idealised case, induction in a realistic 3-D earth with a finite conductivity is much more complex and
there is a significant contribution from the anomalous part of the induced (secondary) field due to conductivity anomalies.
Realistic induction is a diffusive phenomenon. It means that there is always some delay in the formation of the induced
currents and related internal fields after a change in the external field. This can be seen when inspecting the animation provided
as supplementary data of this paper. An extreme example in the time domain is a step-like change in the amplitude of the
external current to which the earth would respond by more slowly decaying induced currents. It would mean that, after the
step change in the external field, $d\boldsymbol{H}/dt$ would solely consist of the internal contribution. In turn, the variation field $\boldsymbol{H}$ would
finally be produced only by the external currents that would remain at the enhanced level.

### 4.3 Sources of uncertainty in the analysis

The resolution of the small-scale structures is limited by the station separation of the magnetometer array. We examine this
effect by performing a test with the station KIR. As can be seen in Fig. 7, it is located in the densest part of the network
and typically has a relatively low induced contribution. By removing the three nearest stations ABK, KIL, and MUO, we
can significantly decrease the density of the network around KIR. We run the magnetic field separation with this reduced
network and then compute $k$, similar to the analysis presented in Sect. 3.2 and 3.3. The resulting internal contributions are 26%
(22%) for $B_x$, 39% (30%) for $B_y$, 58% (47%) for $dB_x/dt$, and 66% (51%) for $dB_y/dt$. The numbers in parenthesis give the
corresponding contribution for the intact network (Table 1). There is some increase in the internal contribution with the reduced
network, indicating that structures smaller than what the network can resolve at 90 km altitude may be mapped underground





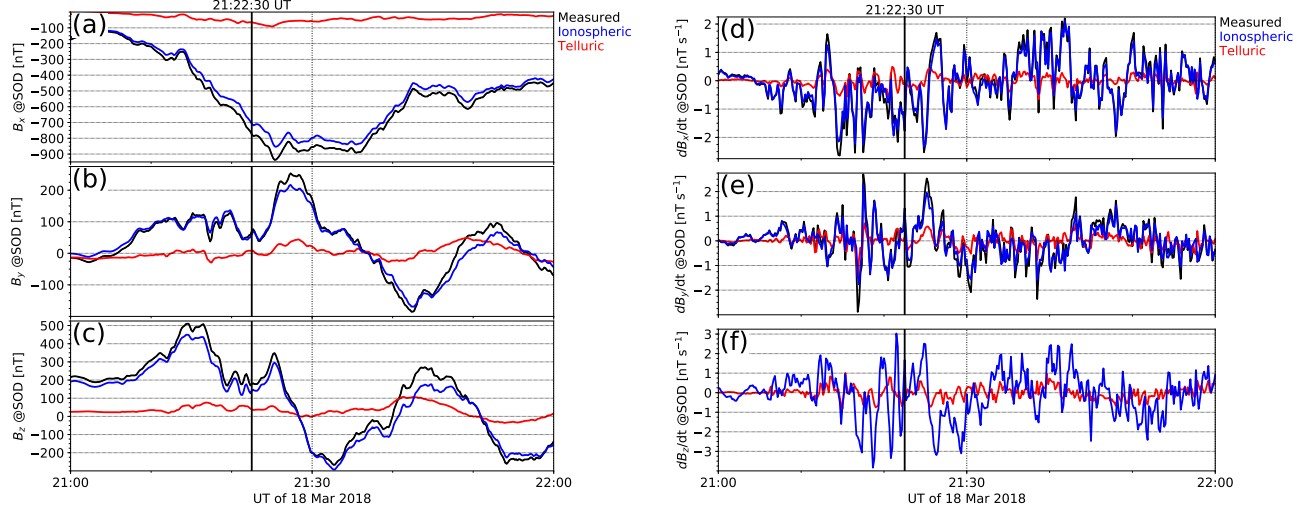

**Figure 9.** Magnetic field at SOD in the same format as Fig. 4 except that the external instead of the real measured ground magnetic field has been given as input to the SECS field separation. The label "Measured" refers to the external magnetic field from the original analysis, the label "Ionospheric" to the external magnetic field from the re-analysis, and the label "Telluric" to the internal magnetic field from the re-analysis.

instead. However, the relative behavior of the different parameters remains unchanged. This indicates that although our numbers are somewhat sensitive to the station configuration, the conclusions drawn from them should still be valid.

Thébault et al. (2006) have shown that perfect separation of the ground magnetic field into internal and external parts is not possible using spherical cap harmonics. The separation should be possible globally, but in a regional case the two sources will be partially mixed, most likely due to boundary conditions, i.e., currents outside of the examined region. Nonetheless, the separation has been considered useful (Stening et al., 2008; Gaya-Piqué et al., 2008). It is likely that the same fundamental problem concerns the regional field separation carried out using the SECS method, and affects our results. We examine this by performing a small test on our example event. We give the separated external (internal) field as input to the SECS method and examine the resulting internal (external) part. For a perfect separation, this should be zero, of course. The results for the external and internal input field are illustrated in Fig. 9 and Fig. 10, respectively.

Fig. 9 and Fig. 10 show that the field separation performed using the SECS method and IMAGE data is not perfect, indeed. The re-analysis of the external field produces a small internal part and, vice versa, re-analysis of the internal field produces a small external part. Nonetheless, both the magnetic field and its time derivative are strongly dominated by the field contribution used as input, indicating that although our numbers must be affected by the imperfect field separation, the conclusions drawn from them should still apply.





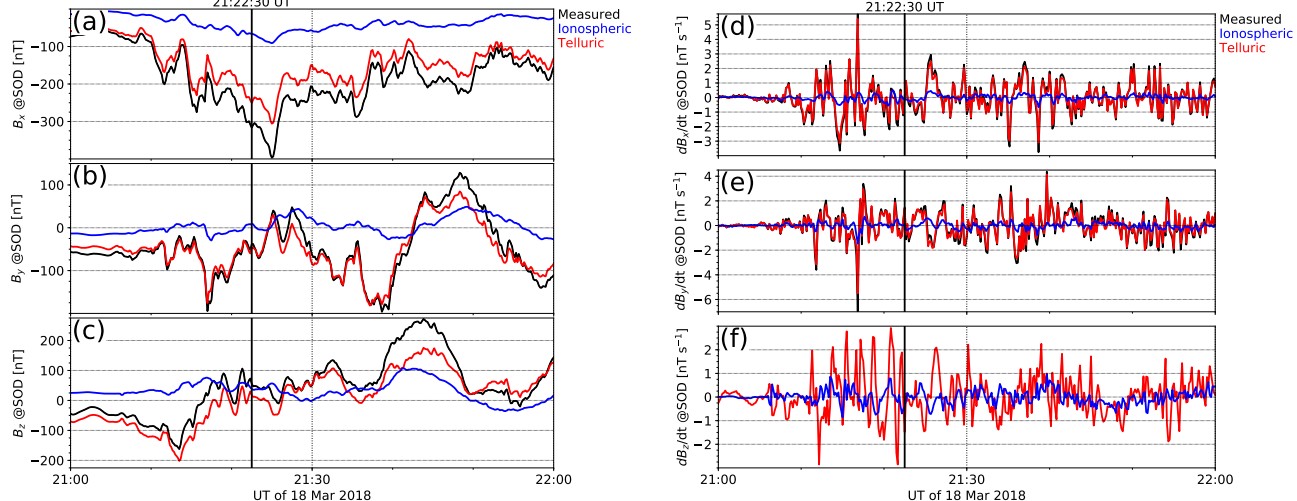

**Figure 10.** Magnetic field at SOD in the same format as Fig. 4 except that the internal instead of the real measured ground magnetic field has been given as input to the SECS field separation. The label "Measured" refers to the internal magnetic field from the original analysis, the label "Ionospheric" to the external magnetic field from the re-analysis, and the label "Telluric" to the internal magnetic field from the re-analysis.

### 4.4 Implications of the results

The significant role of the induced component to the time derivative of the ground magnetic field has some interesting implications. First of all, observations of the time derivative should be considered highly local, and any results derived from them should not be generalized to other locations without caution. It is well known that the electric field at the Earth's surface is highly local and 3-D conductivity structures strongly affect its variability (Kelbert, 2020). When comparing simultaneous measured time derivative values at different locations, it should be kept in mind that they do not necessarily provide a comparable measure of the dynamics of the driving ionospheric currents because they are affected by the internal anomalous fields. Second, attempts to predict the time derivative of the ground magnetic field using global simulations have not been considered very successful (Pulkkinen et al., 2013). According to our results, one significant source of difference between the simulated and measured values is that the simulations typically do not include a conducting ground. Thus, while the simulated magnetic field time derivatives mainly represent the ionospheric currents, the measurements with which they are compared may be dominated by the telluric currents. Lately, there have been some studies where a 3-D conducting ground has been included in a magnetohydrodynamic (MHD) simulation (e.g., Honkonen et al., 2018; Ivannikova et al., 2018).

Separating the magnetic field into telluric and ionospheric parts has the effect that the ionospheric equivalent current density time derivative patterns become less broken than deriving them without the field separation. However, the lifetimes of the ionospheric structures are still very short, comparable with the 80–100 s limit derived by Pulkkinen et al. (2006) for pre-





dictable behavior of the measured ground magnetic field time derivatives. Thus, learning to predict the occurrence of large time derivatives of the ground magnetic field still requires more work.

From the GIC modelling viewpoint, the (horizontal) geoelectric field is the primary quantity as it is the driver of induced currents in technological conductors. While the internal contribution to the magnetic field is only produced by telluric currents,

due to the inductive nature of the magnetic field, the electric field is affected by galvanic effects as well, due to charge accumulation across lateral conductivity gradients. This adds a lot of spatial complexity to the electric field compared to the magnetic field (e.g., Lucas et al., 2020), and is responsible for the strong amplification of the electric field on the less conductive side of a conductivity contrast (e.g., the coast effect). The behavior of $dH/dt$ falls between the rather smoothly varying magnetic field and the spatially very unhomogeneous electric field.

## 5 Conclusions

We have examined the relative contribution of the telluric (secondary, induced) and ionospheric (primary, inducing) electric currents to the variation magnetic field measured on the ground in the time domain. We have used $10 \, \mathrm{s}$ data from the North European IMAGE magnetometer network in 1994–2018, and separated the measured field into telluric and ionospheric parts using the 2-D SECS method. Only relatively large horizontal time derivative values ($> 1 \, \mathrm{nTs}^{-1}$) have been included in the

analysis. Our main results are:

1. Time derivative of the measured horizontal magnetic field ($d\boldsymbol{H}/dt$) is typically dominated by the contribution from the secondary telluric currents.

2. Unlike its time derivative, the horizontal magnetic field ($\boldsymbol{H}$) is typically dominated by the primary ionospheric currents in the vicinity of the source currents.

3. The coast as well as conductivity anomalies (Rikitake and Honkura, 1985; Korja et al., 2002; Engels et al., 2002) tend to rotate $d\boldsymbol{H}/dt$ and increase the internal contribution at nearby stations.

4. We suggest that $d\boldsymbol{H}/dt$ is typically dominated by induced currents and $\boldsymbol{H}$ by ionospheric currents, because shorter periods are more pronounced in $d\boldsymbol{H}/dt$ than in $\boldsymbol{H}$, and their signature is strongly amplified by the Earth.

Our results have been derived using IMAGE data and are thus only valid for IMAGE stations. Some uncertainty in the

numbers is caused by the imperfect separation of the magnetic field into telluric and ionospheric parts due to the spatial resolution of the magnetometer network and boundary conditions. However, it is likely that the main principles, although not the exact numbers, apply, and are relevant to other areas as well.

Our results imply that measurements of $d\boldsymbol{H}/dt$ depend strongly on location, and field separation should be carried out before interpreting them in terms of dynamics of the ionospheric currents. This concerns comparison with simulations as well: either

a 3-D conducting ground should be included in the simulation or the induced part should be removed from the measurements before the comparison. The latter option is obviously preferable if a dense enough measurement network is available, since then no assumptions of the ground conductivity are needed, and computations are much faster.



A natural next step of this study would be to apply a 3-D ground conductivity model together with a given external (equivalent) ionospheric current system in the time domain, and to calculate the external and internal parts of the ground magnetic field and their time derivatives. The approach could be as in Rosenqvist and Hall (2019), with the extension that instead of the frequency domain, the simulation would be performed in the time domain, and the external source would be described by data-based equivalent currents. Such a fully controlled model would provide deeper understanding of the empirical results presented in this study, but would be affected by the limited knowledge of the conductivity structure in Fennoscandia. Improving the conductivity model, in turn, requires much more ground measurements.

*Code and data availability.* IMAGE data are available at https://space.fmi.fi/image. The code used to calculate magnetic coordinates and local times is available at https://apexpy.readthedocs.io/en/latest/. The code used to calculate the wavelet transforms is available at https://pywavelets.readthedocs.io/en/latest/. SMAP data are available on request from Maxim Smirnov (maxim.smirnov@ltu.se) or via the EPOS portal (https://www.epos-ip.org/).

*Video supplement.* IMAGE_20180318T210000_10sec_20180318T220000.mp4 illustrates the time development of the ionospheric and telluric equivalent currents, their time derivatives, and corresponding horizontal ground magnetic fields on 18 March 2018 from 21:00:00 to 22:00:00 UT with a 10 s time step. The animation consists of frames similar to Fig. 3 and Fig. 2.

*Author contributions.* L. J. prepared most of the material and wrote the manuscript. H. V. provided expertise on the theoretical discussion and participated in writing the manuscript. A. V. provided expertise on the GIC application of the results and participated in writing the manuscript. M. S. provided expertise on the MT viewpoint, prepared the conductance maps, and participated in writing the manuscript.

*Competing interests.* The authors declare that they have no conflict of interest.

*Acknowledgements.* This work was supported by the Academy of Finland grant no. 314670. We thank the institutes that maintain the IMAGE Magnetometer Array: Tromsø Geophysical Observatory of UiT the Arctic University of Norway (Norway), Finnish Meteorological Institute (Finland), Institute of Geophysics Polish Academy of Sciences (Poland), GFZ German Research Centre for Geosciences (Germany), Geological Survey of Sweden (Sweden), Swedish Institute of Space Physics (Sweden), Sodankylä Geophysical Observatory of the University of Oulu (Finland), and Polar Geophysical Institute (Russia).





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



**Table 1.** IMAGE station, start and possible end year of operation, number of 10 s data points $N$ with $dH/dt > 1$ nTs$^{-1}$ in 1994–2018, $k$ from $B_{x,telluric} = k \cdot B_x + const.$ for $B_x$, $B_y$, $dB_x/dt$, and $dB_y/dt$.

| Station | Start – end | $N$ | $k_{B_x}$ | $k_{B_y}$ | $k_{dB_x/dt}$ | $k_{dBy/dt}$ |
|---|---|---|---|---|---|---|
| NAL | 1993– | 329135 | 0.69 | 0.79 | 0.72 | 0.80 |
| LYR | 1993– | 659009 | 0.51 | 0.70 | 0.53 | 0.80 |
| HOR | 1993– | 1085398 | 0.48 | 0.68 | 0.64 | 0.79 |
| HOP | 1993– | 785394 | 0.59 | 0.65 | 0.80 | 0.78 |
| BJN | 1993– | 1034847 | 0.57 | 0.65 | 0.78 | 0.79 |
| NOR | 2007– | 251132 | 0.32 | 0.40 | 0.47 | 0.58 |
| SOR | 1982– | 903197 | 0.34 | 0.38 | 0.58 | 0.57 |
| KEV | 1982– | 801550 | 0.32 | 0.46 | 0.54 | 0.68 |
| TRO | 1993– | 1247767 | 0.30 | 0.45 | 0.56 | 0.65 |
| MAS | 1991– | 850303 | 0.27 | 0.36 | 0.60 | 0.60 |
| AND | 1996– | 844389 | 0.33 | 0.56 | 0.55 | 0.71 |
| KIL | 1983– | 982490 | 0.24 | 0.35 | 0.51 | 0.52 |
| IVA | 2001– | 614043 | 0.27 | 0.47 | 0.56 | 0.70 |
| ABK | 1998– | 811199 | 0.24 | 0.39 | 0.45 | 0.54 |
| LEK | 2000–2005 | 275998 | 0.31 | 0.60 | 0.54 | 0.69 |
| MUO | 1982– | 666255 | 0.25 | 0.35 | 0.50 | 0.56 |
| KIR | 1996– | 402716 | 0.22 | 0.30 | 0.47 | 0.51 |
| SOD | 1996– | 525082 | 0.29 | 0.46 | 0.54 | 0.65 |
| PEL | 1982– | 649540 | 0.28 | 0.40 | 0.54 | 0.64 |
| JCK | 2010– | 141634 | 0.25 | 0.40 | 0.58 | 0.65 |
| DON | 2007– | 180815 | 0.32 | 0.52 | 0.69 | 0.68 |
| RAN | 2014– | 46800 | 0.28 | 0.47 | 0.57 | 0.67 |
| RVK | 1999– | 308756 | 0.37 | 0.62 | 0.69 | 0.78 |
| LYC | 1998– | 186200 | 0.35 | 0.51 | 0.69 | 0.74 |
| OUJ | 1992– | 223679 | 0.39 | 0.59 | 0.66 | 0.76 |
| MEK | 2004– | 21266 | 0.38 | 0.69 | 0.61 | 0.79 |
| HAN | 1992– | 73847 | 0.35 | 0.64 | 0.55 | 0.77 |
| DOB | 2000– | 73264 | 0.42 | 0.61 | 0.61 | 0.78 |
| SOL | 2007– | 8631 | 0.43 | 0.56 | 0.60 | 0.71 |
| NUR | 1992– | 51086 | 0.41 | 0.58 | 0.67 | 0.69 |
| UPS | 1998– | 29804 | 0.46 | 0.42 | 0.66 | 0.64 |
| KAR | 2004– | 7186 | 0.48 | 0.67 | 0.61 | 0.80 |
| TAR | 2001– | 12440 | 0.53 | 0.70 | 0.68 | 0.68 |