# Peer review of "Induced currents due to 3D ground conductivity play a major role in the interpretation of geomagnetic variations"

_Annales Geophysicae, 2020_

## Referee Comment (RC1) · Anonymous Referee #1 · 24 May 2020

This manuscript has been meticulously prepared with clear and concise text and well chosen figures that lead the reader through the authors' starting premise, data, analysis and conclusions logically and clearly. While data from the IMAGE array is internationally well known, the approach taken in this paper to analysing it for induction contributions to measured magnetic-field variations is novel and useful.

The paper length is appropriate and the language it uses is fluent and precise. It is a pleasure to read. The title and abstract fit the paper well, the figures are suitable and well described in captions and referenced in the text. Adequate referencing of others' work is evident throughout the paper. Each author's contribution is suitably described.

I can find no fault with this paper, no scientific issues or technical corrections, and recommend its publication as is.

---

## Referee Comment (RC2) · J. Miquel Torta (Referee) · 1 Jun 2020

Juusola and co-authors present evidence from the use of 25 years of 10s data from the IMAGE magnetometer network of the importance of separating external and internal fields when interpreting the measured geomagnetic variations, or when attempting to forecast them. Forecasting the field's time-derivative is certainly one of the current challenges in GIC research, and this paper shows how difficult it is, due to the complexity of the 3-D distribution of the electrical conductivity of the Earth. The authors do an admirable job of proving these facts with, first, an example event and, second, with several seemingly robust results from an impressive statistical sample of spatial models in Scandinavia every 10 s, provided dH/dt > 1 nT/s. Since, in addition, the manuscript is clearly and concisely written, clearly laid out and well presented, it has an important

educational value. The abstract is succinct and comprehensible; the manuscript is logically organized, and adequately illustrated; figures are understandable and readable; English usage and grammar is adequate.

I, therefore, believe this manuscript should be accepted for publication in Annales Geophysicae with minor revisions, or at least a response to the following noted concerns. Perhaps the authors could incorporate some additional references (provided throughout this review) that escaped their state-of-the-art examination of the issues at hand. I would like to stress that I am co-author of some of these papers.

- When I saw the title of the paper before reading it, I thought that it would mainly deal with the importance of properly separating internal from external fields when using geomagnetic variations to study the dynamics of the overhead current systems that originate them. However, as I emphasized above, and the authors acknowledge in their abstract, the results presented in this work show also how important is to accurately taking the 3-D distribution of the electrical conductivity of the Earth into account when attempting to predict the geoelectric field derived from the geomagnetic variations. This is of great importance today and, although I agree that this is not in contradiction with the concept of 'the interpretation of geomagnetic variations' in general, I think that the latter fact should be better recognized in the title and acknowledged in the final conclusions.

- l. 21-24. Neglecting the internal part and interpreting the ground field only in terms of ionospheric (and magnetospheric) equivalent currents has been common in space physics, not only because the typical internal contribution is only of about 10–30%, but also because, in general, the real and modeled separated fields are approximately in phase. When this occurs, the analyses can still afford reliable information about the dynamics of the overhead current systems.

- l. 39-50. A more recent reference on the importance of the effects in areas of sharp lateral conductivity gradients at ocean-land boundaries should be added (e.g., Gilbert,
2005, 2014, Pirjola, 2013).

Gilbert, J. L. (2005), Modeling the effect of the ocean-land interface on induced electric fields during geomagnetic storms, Sp. Weather, 2, 20–28, doi:10.1029/2004SW000120.

Gilbert, J. L. (2014), Simplified Techniques for Treating the Ocean-Land Interface for Geomagnetically Induced Electric Fields, IEEE Int. Symp. Electromagn. Compat., (6), 566–569.

Pirjola, R. J. (2013), Practical Model Applicable to Investigating the Coast Effect on the Geoelectric Field in Connection with Studies of Geomagnetically Induced Currents, Adv. Appl. Phys., 1(1), 9–28.

- l. 86. 'all analyses in this study are carried out in the time domain'

- l. 93. 'Because most IMAGE stations are variometers without absolute references to compensate for any artificial drift, . . .'

- l. 99. The geomagnetic field continuation method of spherical elementary current systems (SECS) is ubiquitous, and now nearly "traditional", but the authors only cite Finnish papers when referring to it. Please, at least put 'e.g.' at the beginning of those citations.

- l. 101-104. With the poles of the elementary currents separated by the order of 100 km, and the measuring stations much more, placing the internal current sheet at only 1 m depth is, in my opinion, exaggerated. In the worst case (sea water conductivity and frequency 1 Hz) the skin depth is a few hundred meters. A few tens of km is probably a more reasonable value.

A problem with putting the elementary currents so close to the surface is that the vertical component of the field (Z) shoots up as one approaches them, and I wonder if this can get one into trouble when synthesizing the surface magnetic field from the model close to one of the SECS poles. Another question that comes to my mind related with

this is: does the external-internal separation not depend on the depth at which these currents are placed if it can be so variable?

- l. 106-108. It is recognized that data gaps forced the waste of usable data. A way to deal with this inconvenience would consist in introducing a temporal dependence in the SECS formulation, in the way of Marsal et al. (under revision in Space Weather – AGU, https://www.essoar.org/doi/10.1002/essoar.10502437.1). This would probably provide smoother time derivatives than analyzing the data at snapshots, because the combined spatial-temporal inversion (using either singular value decomposition or regularized least squares) tends to better absorbing local artificial time-derivative peaks in the data.

- l. 154. Can you explain why the ionospheric and telluric Bz do not appear to be oppositely directed for approximately the last 20 minutes of Figure 4?

- l. 287-294. Yes, the external-internal separation is a problem inherent to any regional technique. However, the separation is better when the area of existence of the geomagnetic field variation is to some extent coincident with the region defining the analysis, or when some regional part of the global source field can be separated, because of its independence or symmetry, from the remainder of the variation source (see Torta and De Santis, 1996; Torta, 2020). Therefore, it would be desirable for the region with measurements to fully include the region in which auroral currents are confined. The effects of the uneven spatial distribution of magnetic data within the entire auroral cap could perhaps be reasonably avoided by SECS if the elementary currents were also spaced at varying densities (see Marsal et al., 2017). I would like to see more discussion about these facts in the paper.

Marsal, S., J. M. Torta, A. Segarra, and T. Araki (2017), Use of Spherical Elementary Currents to map the polar current systems associated with the geomagnetic sudden commencements on 2013 and 2015 St. Patrick's Day storms, J. Geophys. Res. Space Physics, 122, 194–211, doi:10.1002/2016JA023166.

Torta, J.M. (2020), Modelling by Spherical Cap Harmonic Analysis: A Literature Re-

view, Surv Geophys 41, 201–247. https://doi.org/10.1007/s10712-019-09576-2

Torta J.M., De Santis A. (1996), On the derivation of the earth's conductivity structure by means of spherical cap harmonic analysis. Geophys J Int 127, 441–451.

l. 314-315. The meaning of the sentence 'Separating the magnetic field into telluric and ionospheric parts has the effect that the ionospheric equivalent current density time derivative patterns become less broken than deriving them without the field separation' is not clear. In any case, can you give a physical or mathematical explanation for this fact?

Best wishes,

J.M. Torta
* * *

---

## Author Comment (AC1) · 10 Jul 2020

We thank the reviewer for their comments. It's always a pleasure to receive such a review report.

---

## Author Comment (AC2) · 10 Jul 2020

**Response to reviewer #2**
angeo-2020-21
Induced telluric currents play a major role in the interpretation of geomagnetic variations
Liisa Juusola, Heikki Vanhamäki, Ari Viljanen and Maxim Smirnov

Thank you for your comments and suggestions regarding our manuscript. You'll find our replies below. We hope that these answers and the proposed changes to the manuscript are satisfactory.

Sincerely yours,
Heikki Vanhamäki, Ari Viljanen and Maxim Smirnov
(Liisa Juusola is presently on maternity leave)

1) When I saw the title of the paper before reading it, I thought that it would mainly deal with the importance of properly separating internal from external fields when using geomagnetic variations to study the dynamics of the overhead current systems that originate them. However, as I emphasized above, and the authors acknowledge in their abstract, the results presented in this work show also how important is to accurately taking the 3-D distribution of the electrical conductivity of the Earth into account when attempting to predict the geoelectric field derived from the geomagnetic variations. This is of great importance today and, although I agree that this is not in contradiction with the concept of 'the interpretation of geomagnetic variations' in general, I think that the latter fact should be better recognized in the title and acknowledged in the final conclusions.

Thank you for pointing this out, understanding geoelectric fields was indeed one important motivation for this study. Based on your comment we intend to change the title to "Induced currents due to 3D ground conductivity play a major role in the interpretation of geomagnetic variations". Additionally after line 347 we will add a comment "On the other hand, the local amplification of short period dH/dt indicates that the 3-D distribution of the electrical conductivity of the Earth has a major effect on the induced currents and electric fields. Therefore, if simulations are used to predict the geoelectric field or GIC, 3-D induction modeling should be used".

2) l. 21-24. Neglecting the internal part and interpreting the ground field only in terms of ionospheric (and magnetospheric) equivalent currents has been common in space physics, not only because the typical internal contribution is only of about

10–30%, but also because, in general, the real and modeled separated fields are approximately in phase. When this occurs, the analyses can still afford reliable information about the dynamics of the overhead current systems.

Thank you, this is an important point to note. Although there is some effect due to distant magnetospheric currents (like the ring current), it is small at high latitudes. We could clarify the text by changing the text on lines 23-24 to "this is often a reasonable assumption at and close to auroral latitudes, since a typical internal contribution is there about 10-30%." and add after that: "Additionally, the external and internal fields are often approximately in phase, in which case the dynamics of the ionospheric current systems can be estimated reliably without carrying out the separation".

3) l. 39-50. A more recent reference on the importance of the effects in areas of sharp lateral conductivity gradients at ocean-land boundaries should be added (e.g., Gilbert, 2005, 2014, Pirjola, 2013).

As we mention in the manuscript, this phenomenon is not only related to coast-land boundary but any sharp conductivity contrasts would produce similar effect. These effects has been studied in magnetotellurics since Parkinson (doi: 10.1111/j.1365-246X.1959.tb05776.x). However, it's good to mention also the recent modeling efforts. We will add Parkinson and the articles you mentioned in lines 39-40. Additionally Dong et al. (doi: 10.1155/2015/761964 seems to be a relevant article, so we'll add it too.

4) l. 86. 'all analyses in this study are carried out in the time domain'

Will fix.

5) l. 93. 'Because most IMAGE stations are variometers without absolute references to compensate for any artificial drift, . . .'

Will add.

6) l. 99. The geomagnetic field continuation method of spherical elementary current systems (SECS) is ubiquitous, and now nearly "traditional", but the authors only cite Finnish papers when referring to it. Please, at least put 'e.g.' at the beginning of those citations.

This is a fair point, although we note that the first 4 articles we mentioned (Amm, 1997; Amm and Viljanen, 1999; Pulkkinen et al., 2003a,b) can be consider as seminal, while the last one (Vanhamäki and Juusola, 2020) is a recent review. Nevertheless we will add "e.g.", and also mention McLay and Beggan (doi:10.5194/angeo-28-1795-2010), Weygand et al. (doi:10.1029/2010JA016177) and Marsal et al. (doi:10.1002/2016JA023166) as further examples.

7) l. 101-104. With the poles of the elementary currents separated by the order of 100 km, and the measuring stations much more, placing the internal current sheet at only 1 m depth is, in my opinion, exaggerated. In the worst case (sea water conductivity and frequency 1 Hz) the skin depth is a few hundred meters. A few tens of km is probably a more reasonable value.

A problem with putting the elementary currents so close to the surface is that the vertical component of the field (Z) shoots up as one approaches them, and I wonder if this can get one into trouble when synthesizing the surface magnetic field from the model close to one of the SECS poles. Another question that comes to my mind related with this is: does the external-internal separation not depend on the depth at which these currents are placed if it can be so variable?

In our opinion the internal sheet must be quite close to the surface. Especially in the case of a highly conducting ocean, a significant fraction of the induced currents is concentrated close to the surface. For example in Figure 5 of Engels et al. (2002) almost all of the ocean current is in the top sheet (0-10 km), even though the frequency is low for our purposes (T=2048 s).

It's true that the vertical magnetic field of a SECS diverges at the pole of the system. So if the internal layer is very close to the surface and some magnetometer happens to be in the immediate vicinity of a SECS pole, the vertical magnetic field at that magnetometer would be likely to be explained mostly by that nearby SECS. In that case the magnitude of that SECS would be very small, so its effect would not spread to surrounding areas and the internal current associated with that SECS would be very small.

We use a fixed analysis grid where the SECS spacing is $0.5°$ in latitude and $1.0°$ in longitude. As it happens, there are 3 magnetometers whose horizontal distance to the closest SECS pole is less than 10 km: LYR 9.1 km, HAN 5.2 km and TAR 2.6 km. The results for these stations presented in Figures 2, 7, 8 and in the supplementary material do not appear in any way anomalous. The telluric dH/dt in Figure 8c at LYR is perpendicular to the ionospheric field, but also other nearby stations show large deviations in directions. Moreover, similar behavior can be seen at several mainland stations during other local times (in the supplementary material), so this is unlike to be associated with the proximity to a SECS pole. Note that TAR is the station closest to a SECS pole, but it shows no strange features in Fig. 8.

As a further check we repeated the analysis of our example event (18 March 2018, 21:00:00–22:00:00 UT) using 10 km depth for the internal current. There were no visually detectable changes in Figure 2 or in the horizontal components shown in Figure 4, while there were small small changes in the vertical component of Figure 4. In general the internal/external separation does depend on the depth of

the internal currents, but not much. A good illustration of this is in Figure 1 of Juusola et al. (2016), where in panels b and c the internal currents are at depths of 0 km (really 1 m, same as in this paper) and 30 km, respectively.

Based on the above discussion, we conclude that the analysis performed in this paper is robust, and a re-analysis using larger (10-30 km) depth of the internal currents would be unlikely to change any of our conclusions. However, in future analysis it would be better to pay closer attention to the singularity in the magnetic field. One possibility would be to sub-divide the SECS into smaller parts around the grid cell, and to impose a lower limit to the distance that is used in the magnetic field equation.

8) l. 106-108. It is recognized that data gaps forced the waste of usable data. A way to deal with this inconvenience would consist in introducing a temporal dependence in the SECS formulation, in the way of Marsal et al. (under revision in Space Weather – AGU, https://www.essoar.org/doi/10.1002/essoar.10502437.1). This would probably provide smoother time derivatives than analyzing the data at snapshots, because the combined spatial-temporal inversion (using either singular value decomposition or regularized least squares) tends to better absorbing local artificial time-derivative peaks in the data.

Adding a temporal dimension in the SECS, either using splines or other methods, could indeed mitigate the effect of data gaps, and also help dealing with variable temporal resolutions in the data (see reply to comment 10). We notice that the article has been accepted for publication (doi:10.1029/2020SW002491).

However, if this leads to smoother time derivatives, as you mention, then we would consider it a serious drawback. At least in GIC applications the fastest time derivatives are the most important, so we do not want to smooth them. Additionally, using splines is a non-linear data transformation (which can not be described by simple transfer function), so in principle it may introduce other frequencies in the signal.

Nevertheless, we will add a comment about this possibility after line 108: "We note that a possible way to mitigate the effect of data gaps, and at the same time enable use of magnetometer data with different temporal resolutions, would be to add temporal dimension to the SECS analysis, as recently demonstrated by Marsal et al. (2020). However, representing temporal changes in terms of splines or similar non-linear functions could lead to smoother time derivatives and/or changes in the frequency content of the signal, which should be avoided for example in GIC-related studies."

9) l. 154. Can you explain why the ionospheric and telluric Bz do not appear to be oppositely directed for approximately the last 20 minutes of Figure 4?

Indeed, it seems that the relationship between the internal and external Bz shown in Figure 4c changes towards the end of the event. Additionally, we note that the internal and external parts of By have sometimes opposite signs, even though for a simple ideal conductor the horizontal fields should have the same sign (as is the case for Bx).

We speculate that the nice behavior of Bx is due to the large-scale and slowly varying electrojet, whereas the other components are affected more by smaller scale ionospheric currents. This is further modified by the variable delays and penetrations depths of different frequency components, so the overall behavior of the By and especially Bz components can be complicated. A detailed explanation would probably require 3D induction modeling with realistic ionospheric driving currents.

10) l. 287-294. Yes, the external-internal separation is a problem inherent to any regional technique. However, the separation is better when the area of existence of the geomagnetic field variation is to some extent coincident with the region defining the analysis, or when some regional part of the global source field can be separated, because of its independence or symmetry, from the remainder of the variation source (see Torta and De Santis, 1996; Torta, 2020). Therefore, it would be desirable for the region with measurements to fully include the region in which auroral currents are confined. The effects of the uneven spatial distribution of magnetic data within the entire auroral cap could perhaps be reasonably avoided by SECS if the elementary currents were also spaced at varying densities (see Marsal et al., 2017). I would like to see more discussion about these facts in the paper.

Thank you for this comment, your discussion about the boundary effects sounds very reasonable. However, in this kind of large statistical study covering 25 years we want to limit ourselves to the IMAGE magnetometer network, where the data has quite uniform structure and quality. If the whole auroral oval were considered, we would need to deal with several different data sources. While the SuperMAG initiative has made this a lot easier, the data quality would still be quite variable, the available magnetometer sites would change a lot and the overall temporal resolution would be limited.

We will add a comment after line 290 describing your suggestion: "...affects our results. The effect of remote currents might be reduced and the separation improved by expanding the analysis region and magnetic input data to cover the whole auroral region, where the most intense ionospheric currents flow (Torta and De Santis, 1996; Torta, 2020). This would lead to uneven spatial distribution of magnetic data over the entire auroral region, but that could be reasonably handled by using variable density in the SECS grid (e.g. Marsal et al., 2017). However, in

this study we limit the analysis to the IMAGE network and examine the effect of imperfect internal/external separation on our results by performing ...".

11) l. 314-315. The meaning of the sentence 'Separating the magnetic field into telluric and ionospheric parts has the effect that the ionospheric equivalent current density time derivative patterns become less broken than deriving them without the field separation' is not clear. In any case, can you give a physical or mathematical explanation for this fact?

We suggest following as a more clear formulation: "When the magnetic field is separated into telluric and ionospheric parts, short period and small scale variations are seen to be amplified by the internal field contribution. Thus the ionospheric equivalent current density and especially its time derivative have a more regular spatiotemporal structure than could be concluded if they were derived without the field separation".

We tried to explain this in Section 4.1. Our reasoning is that the shallow penetration depth of the fastest changes and the 3D conductivity variations in the ground produce small scale structures in the telluric magnetic field. At the moment we do not have a more concrete explanation than the general discussion offered in Section 4.1, and a detailed discussion would probably require numerical 3D induction modeling.

---

## Referee Report (RR1)

The authors have convincingly addressed my comments in their reply.

Concerning the statement "However, representing temporal changes in terms of splines or similar non-linear functions could lead to smoother time derivatives and/or changes in the frequency content of the signal, which should be avoided for example in GIC-related studies", let me just note that in Marsal et al (2020) it is discussed the importance of the inter-knot frequency selection of the spline expansion. A previous knowledge of the largest frequencies of the target phenomena to be modeled is suggested, and we recommend that the corresponding Nyquist frequency can be used as an upper bound for such inter-knot frequency.

My recommendation is to accept the paper for publication.

Best wishes,

J.M. Torta

---

## Author Response (AR2)

**Response to editor and reviewer**
angeo-2020-21
Induced telluric currents play a major role in the interpretation of geomagnetic variations
Liisa Juusola, Heikki Vanhamäki, Ari Viljanen and Maxim Smirnov

We thank the editor and reviewer for the comments. We have modified the article as indicated below.

Sincerely yours,
Heikki Vanhamäki

1) Lines 18-20: "Mathematically, it is possible to fully explain the variation field by two equivalent current systems, one at the ionospheric altitude and another just below the Earth's surface.". Please provide 1-2 appropriate references here to justify your statement.

   We added reference to Haines and Torta (1994) https://doi.org/10.1111/j.1365-246X.1994.tb03981.x, who demonstrate how the internal and external equivalent currents can be constructed using spherical (cap) harmonic analysis.

2) Line 45: please explain the terms "tipper vector" and "induction arrow" in the text since the readers of ANGEO are not very familiar with the terminology of geomagnetic induction.

   We have added a clarification: "This is often represented graphically by using Tipper vectors (or induction arrows), which combine the real and imaginary parts of the transfer function at a particular frequency, so that the real arrows point towards highly conducting regions (see for example Fig. 6 in Engels et al., 2002"

3) Last but not least, I would like to ask you to incorporate in your MS the comment made by the referee regarding the article by Marsal et al. (2020).

   We have added a clarification on our discussion of Marsal et al. (2020) based on the reviewer comment: "These issues can be avoided by careful selection of the inter-knot frequency in the spline expansion, based on previous knowledge of the largest frequencies of the target phenomena, as discussed by Marsal et al. (2020)".